# Preparation and Application of Active Bionanocomposite Films Based on Sago Starch Reinforced with a Combination of TiO_2_ Nanoparticles and *Penganum harmala* Extract for Preserving Chicken Fillets

**DOI:** 10.3390/polym15132889

**Published:** 2023-06-29

**Authors:** Alireza Bagher Abiri, Homa Baghaei, Abdorreza Mohammadi Nafchi

**Affiliations:** 1Department of Food Science and Technology, Damghan Branch, Islamic Azad University, Damghan, Iran; alirezaabiri398@yahoo.com (A.B.A.); baghaei.homa@yahoo.com (H.B.); 2Food Technology Division, School of Industrial Technology, Universiti Sains Malaysia, Penang 11800, Malaysia; 3Green Biopolymer, Coatings & Packaging Cluster, School of Industrial Technology, Universiti Sains Malaysia, Penang 11800, Malaysia

**Keywords:** active packaging, sago starch, *Penganum harmala*, chicken meat, antimicrobila activity

## Abstract

The aim of this study was to develop sago starch-based bionanocomposite films containing TiO_2_ nanoparticles and *Penganum harmala* extract (PE) to increase the shelf life of chicken fillets. First, sago starch films containing different levels of TiO_2_ nanoparticles (1, 3, and 5%) and PE (5, 10, and 15%) were prepared. The barrier properties and antibacterial activity of the films against different bacteria strains were investigated. Then, the produced films were used for the chicken fillets packaging, and the physicochemical and antimicrobial properties of fillets were estimated during 12-day storage at 4 °C. The results showed that the addition of nano TiO_2_ and PE in the films increased the antibacterial activity against gram-positive (*S. aureus*) higher than gram-negative (*E. coli)* bacteria. The water vapor permeability of the films decreased from 2.9 to 1.26 (×10^−11^ g/m·s·Pa) by incorporating both PE and nano TiO_2_. Synergistic effects of PE and nano TiO_2_ significantly decreased the oxygen permeability of the sago starch films from 8.17 to 4.44 (cc.mil/m^2^·day). Application results of bionanocomposite films for chicken fillet storage at 4 °C for 12 days demonstrated that the films have great potential to increase the shelf life of fillets. The total volatile basic nitrogen (TVB-N) of chicken fillets increased from 7.34 to 35.28 after 12 days, whereas samples coated with bionanocomposite films increased from 7.34 to 16.4. For other physicochemical and microbiological properties of chicken fillets, similar improvement was observed during cold storage. It means that the bionanocomposite films could successfully improve the shelf life of the chicken fillets by at least eight days compared to the control sample.

## 1. Introduction

Today, the use of biopolymer-based films instead of petroleum materials has attracted a lot of attention because the packaging is renewable, eco-friendly, non-toxic, and biodegradable [1,2]. Starch is a widely used polysaccharide for making biopolymer packages. It is easily available, inexpensive, and biodegradable. Starch can produce clear films without odor and color. Additionally, its films have low oxygen permeability [3]. Biopolymer-based films can carry functional and bioactive agents. These agents can include antioxidants, antimicrobials, essential oils, herbal extracts, nanoparticles, etc. [4]. 

Chicken meat is one of the most widely consumed meats in the world because it is reasonably priced, low in fat, and a rich source of essential amino acids, minerals, and polyunsaturated fatty acids beneficial to human health [5,6]. The presence of these nutrients, along with high moisture and pH, has led to its spoilage so that the fats and proteins in this food product are easily oxidized, and various microorganisms can grow in it [7]. Meat shelf life is affected by several factors, such as endogenous enzymes, the presence of oxygen, exposure to light, storage temperature, and microorganisms [8]. Active food packaging is one effective solution to increase the safety and quality of perishable food products, develop their shelf life, and protect them from stress and environmental conditions [9].

Nanomaterials, especially metal and metal oxide nanoparticles, are among the widest additives used in food packaging films [10] because these active compounds, by creating functional activity in packaging films, improve the shortcoming of biopolymer-based films, such as mechanical, barrier, and optical properties [11]. Titanium dioxide (TiO_2_) nanoparticles are biocompatible, non-toxic, and have excellent chemical stability, safe production, electrochemical properties, high reactivity, and low cost; and they also show remarkable photocatalytic characteristics and have excellent chemical stability [12]. The effect of TiO_2_ nanoparticles on improving the barrier, optical, mechanical, and antimicrobial properties of various biopolymer-based films has been approved by researchers [13,14].

Herbal extracts and essential oils are good alternatives to synthetic and chemical preservatives because not only do they not have the disadvantage of chemical additives, but also, they have health effects for humans. Bioactive compounds isolated from natural sources often indicate noticeable antioxidant and antimicrobial activity [15]. Espand (*Peganum harmala*) is a member of *Zygophylaceae* family and is widely distributed in the Mediterranean region and is also found in North Africa, Central Asia, Australia, and the United States. The major traditional uses of this herb include gastrointestinal, cardiovascular, endocrine, nervous, pain relieving, neoplasm and tumours, diabetes, disinfectant, antipyretic, respiratory diseases, skin and hair, arthritis ulcers, inflammation, and rheumatism [16]. *Peganum harmala* has different pharmacological effects, including hypothermic, anticancer, antifungal, anti-nociceptive, antibacterial, and reversible monoamine oxidase inhibition [17]. The extracts from the seeds of *P. harmala* contain anthroquinons, β- carbolinalkaloids, and flavonoid glycosides. Previous studies have claimed the presence of alkaloids in *P. harmala* seeds for their biological and pharmacological activities [18]. The antimicrobial and antioxidant activity of *P. harmala* seed extracts has been reported by researchers [19,20]. Based on our knowledge, the synergistic effects of the *P. harmala* seed extracts and nano TiO_2_ were not investigated in polymeric packaging. So, the objectives of the study were to evaluate the effects of combinations of *P. harmala* seed extracts and nano TiO_2_ on the properties of sago starch film and the application of the active film for the preservation of a real food sample. 

## 2. Materials and Methods

### 2.1. Materials

Sago starch and TiO_2_ nanoparticles were purchased from SIM Supply Company Bhd. (Penang, Malaysia) and Sigma Chemical Co. (St. Louis, MO, USA). Glycerol plasticizer was also obtained from Sigma Chemical Co. (St. Louis, MO, USA). Bacterial strains, including *Escherichia coli* O157:H7, *Staphylococcus aureus*, *Listeria monocytogenes*, and *Salmonella enteritidis*, were prepared by the Organization of Scientific and Industrial Research (Tehran, Iran). The culture medium and chemicals used in this research were purchased from Merck Co. (Darmstadt, Germany).

### 2.2. Preparation of Peganum Harmala Extract

First, the seeds of *Peganum harmala* were freshly collected and pulverised by an electric mill after washing with water and drying in the shade. The seed powder (40 g) was added to 80% ethanol (100 mL), and after stirring at room temperature for 24 h, the resulting mixture was filtered through Whatman No. 1 filter paper. Next, the solvent was removed through evaporation using a rotary evaporator set at 40 °C. The resulting extract was kept in a glass jar covered with aluminium foil at a refrigerated temperature (4 °C).

### 2.3. Preparation of Bionanocomposite Films

To prepare homogenous solutions of TiO_2_ nanoparticles with concentrations of 1%, 3%, and 5% (nanoparticles weight to dry starch weight), appropriate amounts of TiO_2_ nanoparticles were poured into distilled water (100 mL) and then homogenized by an ultrasonic bath (for 20 min). After adding 4 g of sago starch and 2 g of glycerol (50% *w*/*w*) to the TiO_2_ nanoparticles mixtures, stirring was performed. Subsequently, the mixture was heated to a temperature of 90 °C and maintained at this level for 30 min to ensure the complete gelatinization of starch [14]. After reducing the temperature of the mixtures to 45 °C, different concentrations of the *Peganum harmala* extract (PE) (5%, 10%, and 15% *v*/*v*) were added to the film solutions. The mixture was then homogenized for about 30 min. Certain film solutions (approximately 90 g) were poured onto Plexiglas plates (16 cm × 16 cm) and kept at room temperature for 24 h. The produced films were placed in a saturated magnesium nitrate desiccator to balance their moisture.

### 2.4. Tests performed on Bionanocomposite Films

#### 2.4.1. Measurement of the Film Thickness 

The thickness of bionanocomposite films was determined at five points of the film using a digital vernier calliper (with 0.001 mm accuracy).

#### 2.4.2. Measurement of the Water Vapor Permeability 

The water vapor permeability (WVP) of film samples was measured using the ASTM E95−96 standard. Initially, the samples were conditioned at 57% RH overnight. The films were glutted on glass vials separately, and the vials were filled with calcium sulfate (3 g) and weighed. They were then placed in a desiccator containing a saturated potassium sulfate solution. The weight of the vials was measured every two hours for a week, through which a graph of weight changes against time was drawn, and the slope of the resulting line was determined. Finally, the WVP was obtained by the following equation where P = the saturated vapor pressure of water at room temperature (Pa), R1 = the RH of the desiccator, R2 = the RH of the vial, and X = the film thickness (m) [21]:$$\mathrm{WVP}=\frac{\mathrm{WVTR}}{P\;\left(R2-R1\right)}\;X\;$$

#### 2.4.3. The Oxygen Permeability

The oxygen permeability (OP) of the films was measured using the ASTM D3985-17 standard and Mocon Oxtran 2/21 system (Brooklyn Park, MN, USA). Initially, the film sample was conditioned for 48 h (at 55% RH, 25 °C) and, after measuring its thickness, mounted into the diffusion cell of the device. The OP of the sample was estimated using the convergent method by WinPermTM permeability software and expressed as cc.mil/m^2^·day [22].

#### 2.4.4. Antibacterial Activity Assays

The antibacterial activity of the film samples against *Escherichia coli*, *Staphylococcus aureus*, *Listeria monocytogenes,* and *Salmonella enteritidis* was investigated, and the agar disc diffusion method and the diameter of the growth inhibition zone around each disc was used for this purpose. The film discs with a diameter of 6 mm were prepared and placed on Muller Hinton Broth medium. Before placing the disc samples, a surface culture was performed using the liquid culture of each of the bacteria strains (0.1 mL). Finally, the plates were incubated at 37 °C for 24 h, and then the growth inhibition zone area was determined using a digital micrometre (with 0.02 mm accuracy) [23].

### 2.5. Preparation of Coated Chicken Fillets with Bionanocomposite Films

The filleted chicken breast was prepared from the slaughter of the day and transferred to the laboratory in completely hygienic conditions. The chicken fillets were chopped into equal-sized species (200 g). The physicochemical and microbial tests were performed on day 1. The fillets were covered with bionanocomposite films and kept in the refrigerator (4 °C) for 12 days. The unpacked fillets were also considered as a control sample. The chicken fillets were examined every 4 days during the storage period.

### 2.6. Tests Performed on Chicken Fillets

#### 2.6.1. Measurement of pH 

To measure the pH of the samples, first, 10 g of each sample was mixed well with 90 mL of distilled water. Then, the pH of the resulting mixture was read by a pH meter at room temperature (23 ± 2 °C) [24].

#### 2.6.2. Measurement of Total Volatile Basic Nitrogen 

The sample (10 g) was dispersed in distilled water (100 mL), and after stirring for 30 min, the mixture was then filtered. A total of 5 mL of magnesium oxide (10 g/L) was added to 5 mL of filtrate, and then the resulting mixture was distilled through the Kjeldahl system. The distillate was absorbed by a 3% boric acid aqueous solution (25 mL) containing a mixture of methyl red and methylene blue color indicators (0.04 mL). After that, the boric acid aqueous solution was titrated with a hydrochloric acid solution (0.01 mol/L). Total volatile basic nitrogen (TVB-N) values of fillets were calculated by the following equation where V = consumption volume of hydrochloric acid and C = concentration of hydrochloric acid [25]:$$\mathrm{TVB}-N\;(\mathrm{mg}/100\;g)=\frac{V\;\times C\;\times 14\times 100}{10}$$

#### 2.6.3. Microbiological Analysis 

The desired dilution was initially prepared from each sample to perform microbial tests, including total aerobic mesophilic bacteria (TAMB), psychrophilic bacteria, coliforms, molds, and yeasts. A total of 10 g of sample was added to 90 mL of 85% sterile physiology serum, mixed well, and homogenized. A total of 100 μL of a diluted sample was spread on pre-prepared medium plates. Culture and counting of TAMB were performed using plate count agar (PCA) medium at 37 °C for 48 h. Psychrophilic bacteria were also determined using PCA medium PCA at 7 °C for 10 days. Coliforms were studied using the pour overlay method and violet red bile glucose agar (VRBG) medium at 37 °C for 24 h. Molds and yeasts were studied using rose bengal chloramphenicol selective agar (RBCA) medium at 25 °C for 5 days. Finally, the numbers of bacterial colonies were expressed as log CFU/g of the sample [26]. 

### 2.7. Statistical Analysis

All parameters were examined with three replications, and the analysis of their mean was performed by one-way analysis of variance (ANOVA) using the SPSS 22.0 software. Differences between samples were expressed using Duncan’s multiple range test at a 5% level (*p* < 0.05).

## 3. Results and Discussions

### 3.1. The Thickness of the Films

The thickness of biopolymer-based films is important because it affects the mechanical and physical properties of packaging films [27]. The effect of adding different combinations of TiO_2_ nanoparticles and PE on the thickness of sago starch-based films is shown in Figure 1. The thickness values of bionanocomposite films ranged from 0.06 to 0.25 mm. The addition of a combination of TiO_2_ nanoparticles and PE led to a significant increase in the thickness of the produced films from 0.1 to 0.2 mm (*p* < 0.05). The increase in film thickness due to the incorporation of TiO_2_ nanoparticles is due to the rise in the film network volume due to the placement of nanoparticles. Despite a slight increase in thickness due to increasing the concentration of the PE, its increase was not statistically significant. This slight increase in the thickness of the films is probably related to the increase in the solids content of the film-forming solutions due to the addition of PE.

The researchers reported that in films containing nanoparticles, the film density increased due to the interaction between the polymer and the nanoparticles, followed by an increase in the thickness and, consequently, the volume of the films [28]. Previous studies have also reported increasing the thickness of biopolymer-based films containing different nanoparticles, essential oils, and herbal extracts [29,30]. In the research conducted by Asdagh et al. [31], the addition of copper oxide nanoparticles, paprika extract, and coconut essential oil to the whey isolate film formulation showed an increase in the thickness of the produced films. Mehdizadeh et al. [32] also found that incorporating pomegranate peel extract caused a slight increase in the thickness of chitosan-starch-based films.

### 3.2. The Water Vapor Permeability of the Films (WVP)

The high amounts of moisture in food products can be an important factor in accelerating the chemical and microbiological spoilage of food products. Therefore, the food industry is trying to reduce the WVP of films as much as possible by using appropriate additives to develop the shelf life of products [33]. In general, the WVP of films depends on two factors: diffusion of water molecules and solubility [34]. The mean values of WVP of the control sago starch film and the films containing combinations of different levels of TiO_2_ nanoparticles and PE are compared in Figure 2 and demonstrated that adding a combination of nanoparticles and PE to the film samples resulted in a significant reduction in the WVP (*p* < 0.05). The WVP of the bionanocomposites also decreased with increasing the concentration of TiO_2_ nanoparticles from 1 to 5% and PE from 5 to 15% in the film formulation; however, the effect of nanoparticles in reducing the WVP values of the films was higher than PE (*p* < 0.05). Due to intermolecular reactions between the chains of starch, nanoparticles, and extract, the empty spaces between the polymer chains are reduced. Through this, the path movement of water molecules becomes more difficult and can be justified by free volume theory [35,36]. The highest WVP value was observed in the control film (2.90 × 10^−11^ g/m·s·Pa), and the lowest amount was for the film containing a combination of 5% TiO_2_ nanoparticles and 15% PE (1.26 × 10^−11^ g/m·s·Pa). Decreased WVP of biopolymer-based films due to the addition of metal oxide nanoparticles was also reported by other researchers [37,38]. Incorporating different herbal extracts such as paprika, rosemary, bell pepper, and thyme extracts into the biopolymer-based films also showed decreased WVP [31,39].

### 3.3. The Oxygen Permeability of the Films (OP)

Oxygen plays a very important role in destructive reactions in food products, including lipid oxidation and microbial growth, and exposure of foods to high levels of oxygen can lead to the development of the spoilage rate of products. The effect of incorporating different combinations of TiO_2_ nanoparticles and PE on the OP of sago starch-based films is shown in Figure 3. By adding the combination of nanoparticles and extract to the starch film formulation, a significant decrease in the OP of the films was observed, and by increasing the concentration of TiO_2_ nanoparticles from 1 to 5%, the OP was significantly reduced (*p* < 0.05). However, at constant levels of TiO_2_ nanoparticles, increasing the levels of PE did not significantly change the OP of the films. The OP values of different starch-based bionanocomposite films were 4.48 to 8.17 cc.mil/m^2^·day. In general, fillers reduce the mobility of chains by reducing the voids in the film matrix and between its chains, thereby reducing the release rate of oxygen molecules and other gases [40]. Other researchers also confirm the reduction in the OP of starch-based films due to the incorporation of nanoparticles [41,42,43]. Talón et al. [39] found that the addition of thyme extract had no significant effect on the OP of starch-based films.

### 3.4. The Antibacterial Activity of the Films

In this study, the antibacterial activity of bionanocomposite films based on sago starch containing different levels of TiO_2_ nanoparticles and PE on major indicator foodborne bacteria, including *Escherichia coli* O157:H7, *Staphylococcus aureus*, *Listeria monocytogenes*, and *Salmonella enteritidis,* was investigated by disc diffusion method, and the results are presented in Table 1. As seen in the table, the control starch film did not show antibacterial activity, but by adding the combination of TiO_2_ nanoparticles and PE in the films and increasing their levels, the antibacterial activity against the mentioned bacteria increased significantly (*p* < 0.05). Therefore, the highest diameter of growth inhibition zone against *Escherichia coli* O157:H7 (100.82 mm^2^), *Staphylococcus aureus* (112.50 mm^2^), *Listeria monocytogenes* (100.14 mm^2^), and *Salmonella enteritidis* (99.14 mm^2^) was observed in the film sample containing 5% TiO_2_ nanoparticles in combination with 15% PE. Generally, the antibacterial activity of TiO_2_ nanoparticles is exerted through at least two mechanisms: (1) Oxygen-reactive species are produced in a process called auto-catalysis that degrades bacterial cell membranes and membrane integrity and interferes with oxidative phosphorylation, resulting in cell death [44]; (2) Photocatalytic properties of TiO_2_ that have a direct effect on killing bacteria. Antibacterial photocatalytic activity is accompanied by lipid peroxidation, which increases membrane fluidity and destroys cell integrity [45]. Harmaline, harmine, harmol, and tetrahydroharmine are major alkaloids identified in *Peganum harmala* that indicate remarkable functional and biological activities [46]. Research has shown that harman, a highly aromatic alkaloid compound, exerts its antibacterial activity by reacting with DNA [47]. The antibacterial activity of *Peganum harmala* may also be related to the high levels of phenolic compounds with strong antibacterial activity [48].

Kadam et al. [49] evaluated the antibacterial effects of TiO_2_ and SiO_2_ nanoparticles on the soy protein isolate and corn zein edible films. Similarly, they demonstrated that using these nanoparticles in biopolymer-based films resulted in increased antibacterial activity and decreased the growth of microorganisms. Other researchers have also reported the antimicrobial activity of metal dioxide nanoparticles [50,51]. Asdagh et al. [31] also observed the antibacterial activity of coconut essential oil and paprika extract in biopolymer-based films and found that the antibacterial activity of the produced films against gram-positive bacteria (*Bacillus cereus*) was higher than gram-negative bacteria (*E. coli*).

### 3.5. The pH of Chicken Fillets

Figure 4 indicates the changes in the mean pH values of different chicken fillets coated with active bionanocomposite films based on sago starch containing a combination of TiO_2_ nanoparticles and PE during a 12-day storage period at 4 °C. On the first day of experiments, the pH value of chicken fillets was 5.37. In all samples, the pH values increased significantly over time (*p* < 0.05) due to the activity of spoilage bacteria and the consumption of proteins and the production of nitrogen-volatile compounds, such as ammonia and triethylamine [52,53]. Since the control sample and the samples coated with sago starch film did not contain preservative additives, they had the highest growth of microorganisms. They showed a higher pH increase during storage than other samples (*p* < 0.05). So, on the last day of refrigerated storage, the highest pH value was for the control sample (7.34), followed by the sample coated with sago starch film (7.00), and the sample coated with a starch film containing the highest level of TiO_2_ nanoparticles in combination with the highest level of PE (ST5PE15 sample) had the lowest pH amount (5.99). The reduction in pH changes in various types of meat during refrigerated storage due to the application of active biopolymer-based films and coatings containing different nanomaterials, essential oils, and herbal extracts has also been confirmed by previous studies [13,54].

### 3.6. The Total Volatile Basic Nitrogen of Chicken Fillets (TVB-N)

The amount of TVB-N in meats is considered an indicator to evaluate the quality of products because due to the activity of enzymes in meat tissue and the growth and activity of spoilage microorganisms, an increase in TVB-N occurred [55]. The changes in the mean pH values of different chicken fillets coated with active bionanocomposite films based on sago starch containing a combination of TiO_2_ nanoparticles and PE during a 12-day storage period at 4 °C are given in Figure 5. Since chicken fillets had a low microbial load on the first day, their TVB-N value was also low on this day (7.34 mg N/100 g). During the storage period, with increasing the activity of bacteria and proteolytic enzymes and increasing the production of ammonia, trimethylamine, and dimethylamine, the TVB-N values in different fillet samples increased (*p* < 0.05). Due to the lack of preservative in the control sample and in the fillets coated with starch film, the highest rate of a TVB-N increase was observed in these samples; so, these two samples had the highest amount of TVB-N on the last day of storage (35.28 and 30.28 mg N/100 g, respectively), and the lowest amount of TVB-N was observed in fillets coated with a bionanocomposite film containing a combination of 5% TiO_2_ nanoparticles and 15% PE (16.40 mg N/100 g). The reduction in TVB-N production in samples coated with active bionanocomposites is due to the antimicrobial activity of TiO_2_ nanoparticles and PE and, therefore, a reduction in the rate of protein degradation. The maximum acceptable value for the TVB-N of meats is 25 mg N/100 g [53]. The results of this study indicated that the TVB-N values of the control sample were higher than the recommended level from the eighth day and the sample coated with sago starch film from the twelfth day. However, TVB-N values were acceptable in samples coated with active bionanocomposite films containing TiO_2_ nanoparticles and PE until the last day of storage. Reducing the TVB-N values of shrimp due to the use of active film based on polyethylene containing cinnamon and rosemary essential oils and also reducing the TVB-N values of lamb meat due to the application of whey protein–cellulose nanofiber films containing TiO_2_ nanoparticles and rosemary essential oil were observed by Dong et al. [56] and Alizadeh-Sani et al. [13], respectively. Azarifar et al. [54] also found that the application of gelatin-carboxymethylcellulose-based films containing chitin nanofiber and *Trachyspermum ammi* essential oil could significantly reduce the rate of TVB-N production in raw meat samples compared to the control sample.

### 3.7. Microbiology of Chicken Fillets

The changes in the numbers of total aerobic mesophilic bacteria in different chicken fillets coated with active bionanocomposite films based on sago starch containing a combination of TiO_2_ nanoparticles and PE during a 12-day storage period at 4 °C are shown in Figure 6. At the beginning of the storage period, the number of mesophilic bacteria in chicken fillets was 3.35 log CFU/g, which indicates the low microbial load of fillets and their freshness. During the 12-day storage period, the number of mesophilic bacteria in all samples increased significantly (*p* < 0.05), and due to the lack of preservatives in the control sample, the highest growth and proliferation of bacteria were observed in this sample. As expected, the highest number of mesophilic bacteria was observed in the control sample on the last day of refrigeration storage (8.76 log CFU/g), followed by fillets coated with sago starch film without additive (8.30 log CFU/g). Since bionanocomposite films containing 5% TiO_2_ nanoparticles combined with 15% PE had the highest antibacterial activity, the lowest number of mesophilic bacteria was observed in fillets coated with this active film (5.00 log CFU/g). The maximum acceptable number of aerobic mesophilic bacteria in meats is 7 log CFU/g [57]. According to the results of this study, the control sample and the sample coated with sago starch film from the eighth day onwards and the sample coated with a film containing a combination of 1% TiO_2_ nanoparticles and 5% PE from the twelfth day onwards had the number of total mesophilic bacteria higher than the allowable limit. In contrast, other samples had less than the maximum recommended number until the last day of storage.

Psychrophilic bacteria are common strains grown in cold-stored food products. These bacteria can break down sugars (glucose) and amino acids under aerobic conditions and continue to function even when stored cold [58]. The changes in the numbers of psychrophilic bacteria in different chicken fillets coated with active bionanocomposite films based on sago starch containing a combination of TiO_2_ nanoparticles and PE during a 12-day storage period at 4 °C are shown in Figure 7. 

The chicken fillets had 3.26 log CFU/g of psychrophilic bacteria at the beginning of the storage period, and over time, the number of these bacteria in all fillet samples increased significantly (*p* < 0.05). The highest growth of psychrophilic bacteria during storage was related to the control sample. On the last day of storage, the control sample had the highest number of psychrophilic bacteria (8.94 log CFU/g), and the lowest number of them was observed in the fillets coated with a film containing the highest level of TiO_2_ nanoparticles in combination with the highest level of PE (ST5PE15) (5.47 log CFU/g).

The results of examining the number of coliforms in different samples of chicken fillets during 12-day refrigeration storage (Figure 8) demonstrated that during refrigeration storage, the number of coliforms in all samples significantly increased (*p* < 0.05), and the highest rate of that was for the control sample. The application of active bionanocomposite films containing a combination of TiO_2_ nanoparticles and PE was able to remarkability reduce the number of coliforms in the fillet samples, and with increasing the concentration of TiO_2_ nanoparticles and PE in the films, their antimicrobial activity increased, and the number of coliforms showed a significant decrease (*p* < 0.05). So, chicken fillets had 2.16 log CFU/g of coliforms at the beginning of the storage period. Their numbers reached 6.98 logs CFU/g and 2.56 log CFU/g in the control sample and the fillets coated with sago starch-based film containing the highest level of TiO_2_ nanoparticles in combination with the highest level of PE (ST5PE15) on the last day of storage, respectively.

The figure of changes in the average number of molds and yeasts in chicken fillet samples coated with different active bionanocomposite films based on sago starch containing TiO_2_ nanoparticles in combination with PE during refrigeration storage (Figure 9) showed that at the beginning of the storage period the number of molds and yeasts in chicken fillets was 1.62 log CFU/g. Over time, the number of these microorganisms in all samples significantly increased (*p* < 0.05). Similar to other microorganisms studied in this research, the highest number of molds and yeasts in all days of refrigeration storage was for the control sample. On the last day of storage, the number of molds and yeasts in the control sample was 5.87 log CFU/g and the fillets coated with bionanocomposite film containing a combination of 5% TiO_2_ nanoparticles, and 15% PE had the lowest number (3.01 log CFU/g).

Metal and metal oxide nanoparticles can interact directly with microbial cells; they stop the transfer of electrons between membranes, penetrate into the cell space, oxidize cellular compounds, or produce various reactive products, including reactive oxygen species, which cause cell damage [59]. Two general mechanisms have been proposed for the antimicrobial action of phenolic compounds, including the chelating of effective metal ions required for the growth of microorganisms and the reactions with extracellular enzymes of bacteria or cell wall proteins [60]. The researchers said that the presence of alkaloids and other phytochemicals, such as polyphenols, are responsible for the biological activity of the *Peganum harmala* seeds [46]. Various mechanisms have been proposed to explain the antimicrobial action of alkaloids. These mechanisms include preventing DNA synthesis, degrading bacterial homeostasis, inhibiting efflux pumps, penetrating the outer membrane, increasing the permeability of the cytoplasmic membrane, and causing leakage of intracellular compounds [61]. The antimicrobial activity of *Peganum harmala* extract against different bacteria has also been observed by Abderrahim et al. [62] and Darabpour et al. [63].

Azarifar et al. [54] similarly showed that during 12-day refrigeration storage, the number of mesophilic bacteria, psychrophilic bacteria, lactic acid bacteria, molds, and yeasts in raw meat samples increased significantly; however, the use of gelatin-CMC-based films containing chitin nanofibers and *Trachyspermum ammi* essential oil could increase the microbial shelf life of meat samples by reducing the growth rate of these microorganisms. A reduction in the microbial load of sea bass fish samples due to the application of active films based on fish protein isolate–fish skin gelatin containing zinc oxide nanoparticles and basil leaf essential oil was reported by Arfat et al. [64]. Alizadeh-Sani et al. [65] also found that coating lamb meat with whey protein isolate film containing TiO_2_ nanoparticles and rosemary essential oil reduced the microbial load of meat samples during the storage period compared to the control sample. Lou et al. [66] also achieved similar results in investigating the effect of LDPE packaging containing TiO_2_ nanoparticles on the microbial shelf life of shrimp.

## 4. Conclusions

The results of this study demonstrated that by incorporating the combination of TiO_2_ nanoparticles and PE into the sago starch-based film formulation, especially at high levels of nanoparticles, the water vapor and oxygen permeability of films were improved. There was a direct relationship between TiO_2_ nanoparticles and PE levels and the antibacterial activity of the films. Increasing the concentration of these two additives in the film samples increased the antibacterial activity against gram-positive and gram-negative bacteria studied in this research. Applying bionanocomposite films, especially at high TiO_2_ nanoparticles and PE levels, could reduce the growth of different microorganisms in chicken fillets. The changes in chemical parameters decreased during the storage period, and the shelf life of the fillets increased. According to the results obtained in this study, it can be concluded that active bionanocomposite films containing a combination of TiO_2_ nanoparticles and PE can be used as active antibacterial packaging to extend the shelf life of meat and meat products. Further research is needed to investigate the toxicity and migration of these active components in order to gather sufficient evidence for their potential industrial application.

## Figures and Tables

**Figure 1 polymers-15-02889-f001:**
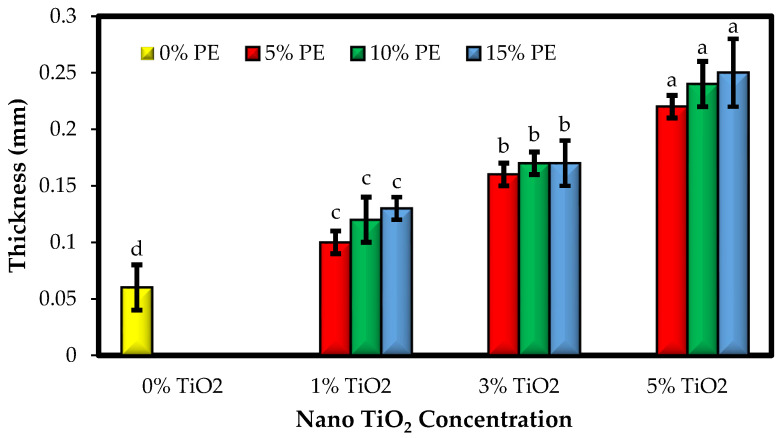
Comparison of the thickness values (mm) of the sago starch/nano TiO_2_/PE bionanocomposite films. Bars represent mean (n = 3) ± SD. Different letters on the bars indicate significant differences among films at a 5% probability level. PE: *Penganum harmala* extract.

**Figure 2 polymers-15-02889-f002:**
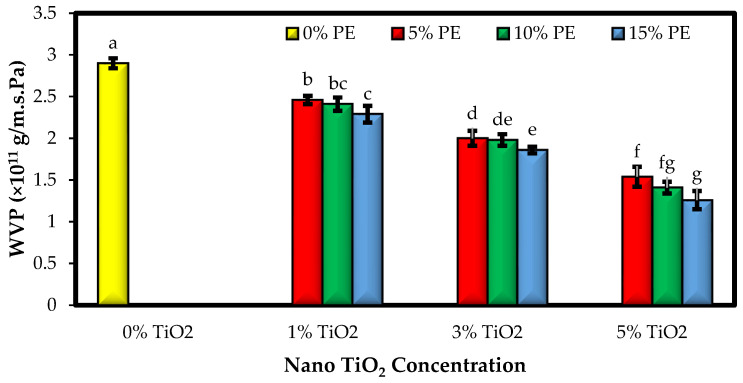
Comparison of the WVP values (g/m·s·Pa) of the sago starch/nano TiO_2_/PE bionanocomposite films. Bars represent mean (n = 3) ± SD. Different letters on the bars indicate significant differences among films at a 5% probability level. PE: *Penganum harmala* extract; WVP: water vapor permeability.

**Figure 3 polymers-15-02889-f003:**
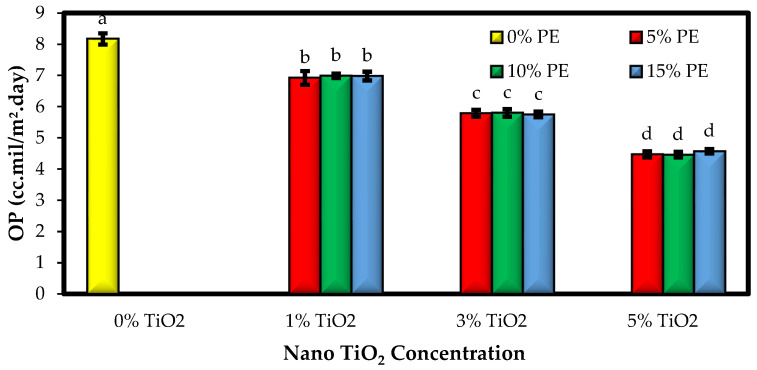
Comparison of the OP values (cc-mil/m^2^. Day) of the sago starch/nano TiO_2_/PE bionanocomposite films. Bars represent mean (n = 3) ± SD. Different letters on the bars indicate significant differences among films at a 5% probability level. PE: *Penganum harmala* extract; OP: oxygen permeability.

**Figure 4 polymers-15-02889-f004:**
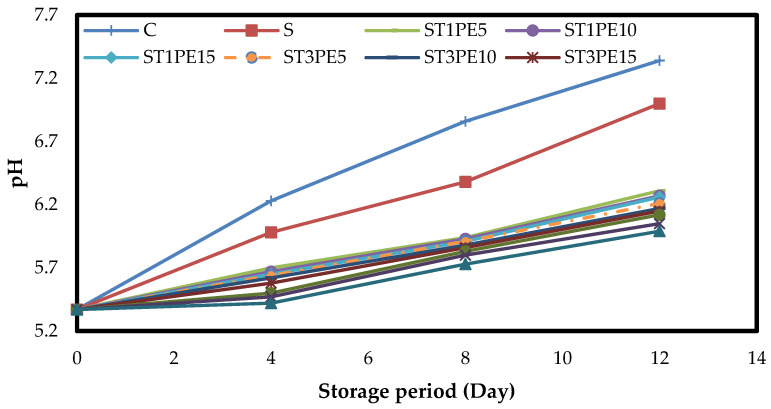
Changes in the pH mean values of different chicken fillet samples coated with sago starch/nano TiO_2_/PE bionanocomposite films during a 12-day storage period at 4 °C. C: control sample; S: fillets coated with sago starch film without additives; T: TiO_2_ nanoparticles; PE: *Penganum harmala* extract.

**Figure 5 polymers-15-02889-f005:**
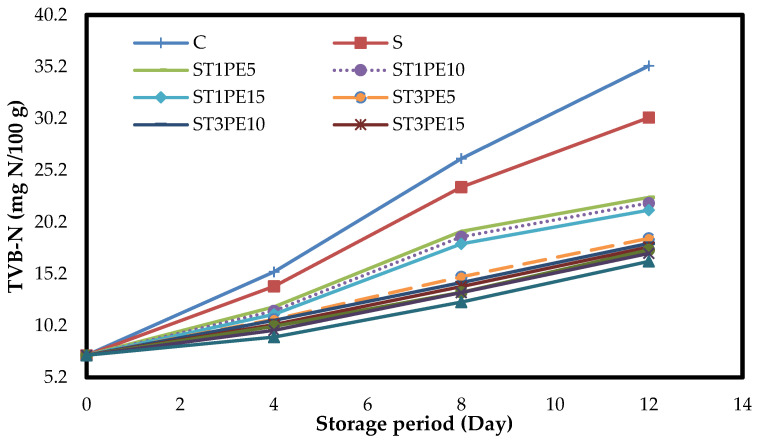
Changes in the TVB-N mean values (mg N/100 g) of different chicken fillet samples coated with sago starch/nano TiO_2_/PE bionanocomposite films during a 12-day storage period at 4 °C. C: control sample; S: fillets coated with sago starch film without additives; T: TiO_2_ nanoparticles; PE: *Penganum harmala* extract; TVB-N: total volatile basic nitrogen.

**Figure 6 polymers-15-02889-f006:**
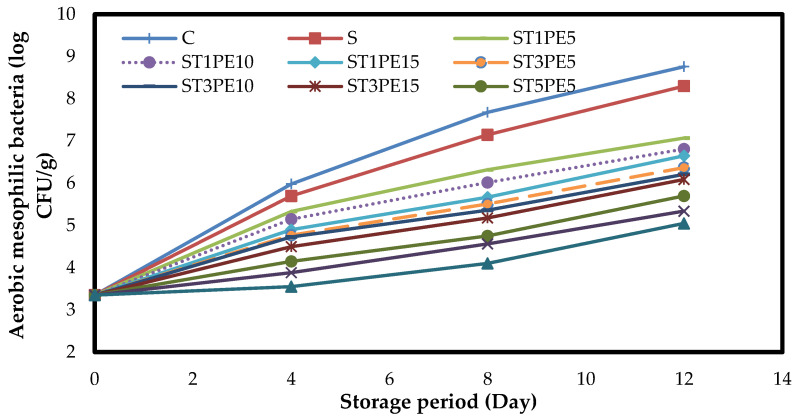
Changes in the numbers of total aerobic mesophilic bacteria (log CFU/g) in different chicken fillet samples coated with sago starch/nano TiO_2_/PE bionanocomposite films during a 12-day storage period at 4 °C. C: control sample; S: fillets coated with sago starch film without additives; T: TiO_2_ nanoparticles; PE: *Penganum harmala* extract.

**Figure 7 polymers-15-02889-f007:**
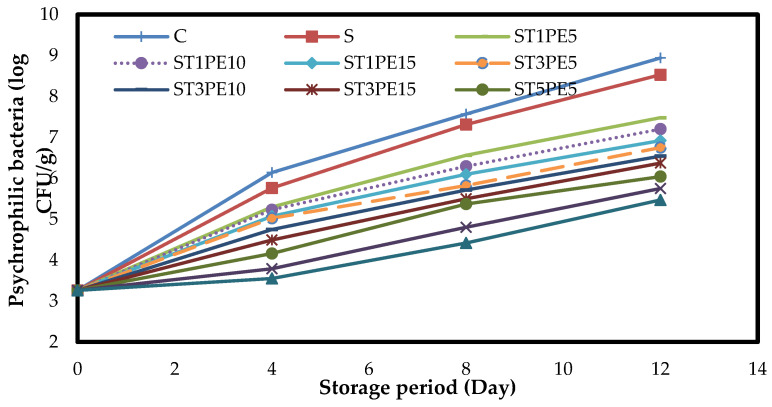
Changes in the numbers of psychrophilic bacteria (log CFU/g) in different chicken fillet samples coated with sago starch/nano TiO_2_/PE bionanocomposite films during a 12-day storage period at 4 °C. C: control sample; S: fillets coated with sago starch film without additives; T: TiO_2_ nanoparticles; PE: *Penganum harmala* extract.

**Figure 8 polymers-15-02889-f008:**
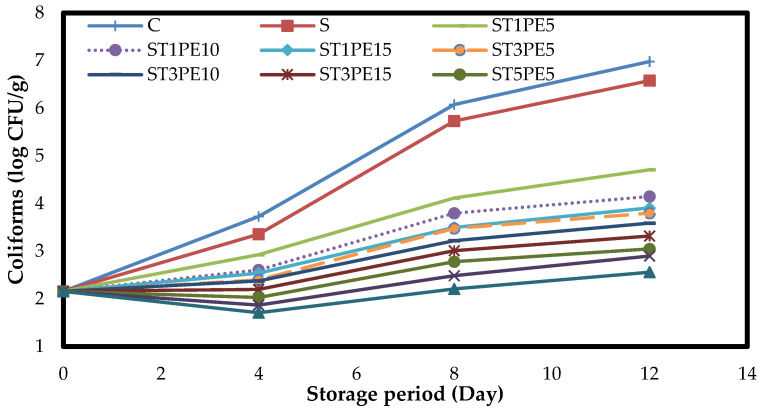
Changes in the numbers of coliforms (log CFU/g) in different chicken fillet samples coated with sago starch/nano TiO_2_/PE bionanocomposite films during a 12-day storage period at 4 °C. C: control sample; S: fillets coated with sago starch film without additives; T: TiO_2_ nanoparticles; PE: *Penganum harmala* extract.

**Figure 9 polymers-15-02889-f009:**
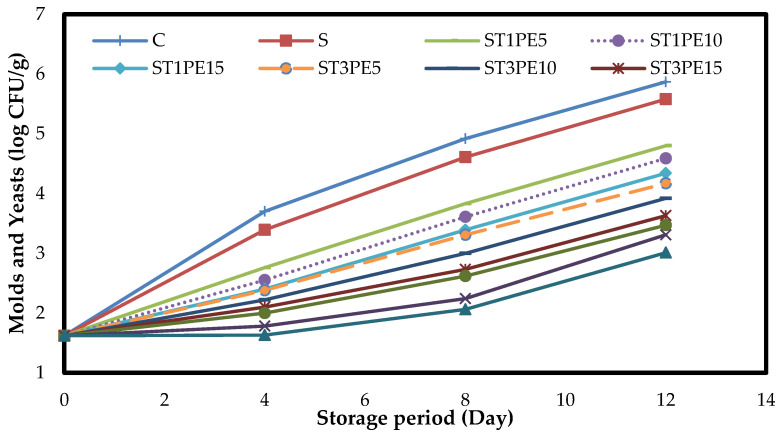
Changes in the numbers of molds and yeasts (log CFU/g) in different chicken fillet samples coated with sago starch/nano TiO_2_/PE bionanocomposite films during a 12-day storage period at 4 °C. C: control sample; S: fillets coated with sago starch film without additives; T: TiO_2_ nanoparticles; PE: *Penganum harmala* extract.

**Table 1 polymers-15-02889-t001:** Comparison of the growth inhibition diameter zone (mm) of the sago starch/nano TiO_2_/PE bionanocomposite films against different bacteria strains.

Film Samples	*E. coli*	*S. aureus*	*L. monocytogenes*	*S. entritidis*
Control	ND	ND	ND	ND
1% TiO_2_ + 5%PE	48.52 ± 0.21 i	63.75 ± 0.25 i	47.32 ± 0.48 i	42.15 ± 0.23 i
1% TiO_2_ + 10%PE	51.74 ± 0.17 h	66.43 ± 0.19 h	50.29 ± 0.34 h	45.17 ± 0.30 h
1% TiO_2_ + 15%PE	56.31 ± 0.32 g	69.49 ± 0.28 g	55.57 ± 0.20 g	49.37 ± 0.16 g
3% TiO_2_ + 5%PE	79.41 ± 0.25 f	92.31 ± 0.45 f	78.31 ± 0.18 f	75.58 ± 0.22 f
3% TiO_2_ + 10%PE	83.18 ± 0.30 e	94.59 ± 0.36 e	81.98 ± 0.41 e	80.33 ± 0.39 e
3% TiO_2_ + 15%PE	86.90 ± 0.09 d	97.13 ± 0.41 d	85.37 ± 0.15 d	84.52 ± 0.17 d
5% TiO_2_ + 5%PE	93.61 ± 0.29 c	107.83 ± 0.33 c	92.82 ± 0.24 c	90.88 ± 0.14 c
5% TiO_2_ + 10%PE	97.59 ± 0.16 b	109.11 ± 0.22 b	96.33 ± 0.19 b	94.07 ± 0.43 b
5% TiO_2_ + 15%PE	100.82 ± 0.41 a	112.50 ± 0.47 a	100.14 ± 0.20 a	99.14 ± 0.25 a

Values represent mean (n = 3) ± SD. Different letters in each column indicate significant differences among films at a 5% probability level. ND: Not Detected.

## Data Availability

The data that support the findings of this study are available from the corresponding author upon reasonable request.

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
