# Peer review of "Preparation and Application of Active Bionanocomposite Films Based on Sago Starch Reinforced with a Combination of TiO2 Nanoparticles and Penganum harmala Extract for Preserving Chicken Fillets"

_polymers, 2023, doi:10.3390/polym15132889_

Round 1
Reviewer 1 Report
The authors have wrote well drafted manuscript entitled "Preparation and application of active bionanocomposite films based on sago starch reinforced with a combination of TiO2 nanoparticles and Penganum harmala extract for preserving chicken fillets". However, their are several correction required as mention below to reconsider the manuscript for publication.
Suggested to rewrite the abstract with addition of actual value, statement like increase or decrease does not reflect the quality investigation.
Line no 34-39: Suggested to break the sentence is very big and complex.
Line no 67. Please when start writing the sentence don't use the short name of plant.
Please correct in whole Manuscript TiO2
Line no 91-92, 100. Suggested to improve the English
Line no 111. Please correct to vernier calliper, if it is digital then mention it too.
Please don't use abbreviations in sub-headings
Line no. 146. suggested write the physical test performed on day 1.
Line no 193. Could you please write the significant value for better understanding of reader
Figure -1, 2 suggested to added significant bar/air-sticks in the figure if their.
Line no 207-208. Please check the statement
Suggested to supplement the zone of inhibition figure, also wether the bor diameter was subtracted?
Suggested to provide toxicity aspect of the film
Also the migration efficacy of TiO2 within the fillets. (Example to test the nanoparticles migration can be found in this paper: https://doi.org/10.1007/s12649-020-01237-5)
Thanks and Good Luck

Moderate editing of English language required
Author Response
Reviewer #1:
The authors have wrote well drafted manuscript entitled "Preparation and application of active bionanocomposite films based on sago starch reinforced with a combination of TiO2 nanoparticles and Penganum harmala extract for preserving chicken fillets". However, their are several correction required as mention below to reconsider the manuscript for publication.
Author’s Reply: Dear reviewer, thanks for your positive feedback. We tried our best to implement all your comments.
Reviewer’s Comment: Suggested to rewrite the abstract with addition of actual value, statement like increase or decrease does not reflect the quality investigation.
Author’s Reply: Thanks for great suggestion. The abstract was revised based on your comments.
Reviewer’s Comment: Line no 34-39: Suggested to break the sentence is very big and complex.
Author’s Reply: Thank you for the comment. Revised.
Reviewer’s Comment: Line no 67. Please when start writing the sentence don't use the short name of plant.
Author’s Reply: Thanks for great comment. Revised.
Reviewer’s Comment: Please correct in whole Manuscript TiO2
Author’s Reply: Thanks for this comment. Revised in whole manuscript.
Reviewer’s Comment: Line no 91-92, 100. Suggested to improve the English
Author’s Reply: Thanks for this comment. Not only these sentences improved, but whole manuscript double checked by one of our professional colleagues in English language.
Reviewer’s Comment: Line no 111. Please correct to vernier calliper, if it is digital then mention it too.
Author’s Reply: Thanks for the comment. It was corrected.
Reviewer’s Comment: Please don't use abbreviations in sub-headings
Author’s Reply: Thanks for this comment. All checked and corrected.
Reviewer’s Comment: Line no. 146. suggested write the physical test performed on day 1.
Author’s Reply: Thanks for the comment. Revised.
Reviewer’s Comment: Line no 193. Could you please write the significant value for better understanding of reader
Author’s Reply: Dear reviewer, thank you. Revised.
Reviewer’s Comment: Figure -1, 2 suggested to added significant bar/air-sticks in the figure if their.
Author’s Reply: Dear Reviewer, really thanks for your great idea. The figures improved.
Reviewer’s Comment: Line no 207-208. Please check the statement
Author’s Reply: Thanks for the comment. The statement revised.
Reviewer’s Comment: Suggested to supplement the zone of inhibition figure, also wether the bor diameter was subtracted?
Author’s Reply: Dear reviewer, unfortunately we don’t have a picture of the inhibition zone to provide as supplementary but will consider for our next research to have a picture of some important phases.
Reviewer’s Comment: Suggested to provide toxicity aspect of the film
Author’s Reply: Dear reviewer, we agree with you. But as TiO2 and the PE extract are GRAS in the range of usage we didn’t test the toxicity. To clarify this, we have added a statement to the conclusion as future research prospects.
Reviewer’s Comment: Also the migration efficacy of TiO2 within the fillets. (Example to test the nanoparticles migration can be found in this paper: https://doi.org/10.1007/s12649-020-01237-5)
Author’s Reply: Nice idea. Unfortunately, this research has already finished, but for sure we will consider for our next research design. To clarify this, we have added a statement to the conclusion as future research prospects.
Thanks and Good Luck
Reviewer 2 Report
Dear Author
thank you for your work
I have only one important question
what is the role and mechanism of using TiO2 in helping all your examined properties? antibacterial is known but the mechanism of the other properties are not, please provide and discuss with your data
Author Response
Reviewer #2:
Dear Author
thank you for your work
I have only one important question
Reviewer’s Comment: what is the role and mechanism of using TiO2 in helping all your examined properties? antibacterial is known but the mechanism of the other properties are not, please provide and discuss with your data
Author’s Reply: Dear Reviewer, thanks for your great comment. TiO2 nanoparticles in other properties of films play a filler role and improve the barrier properties or other physicochemical properties. To clarify this, we have discussed it in lines 230-232 and 256-259.
Reviewer 3 Report
The current work presents systematic data on novel polymer packaging compositions. The work has merit and needs only a few minor updates.
L 190 please add a reference
L 207 intramolecular is likely incorrect as this means that only one type of molecule is involved. Later one intermolecular is used for two starch molecules. Intramolecular is vs additive but this is also intermolecular between starch molecule and additive functional group.
Ref 34. Please refer also to Free Volume Theory and its application in polymer engineering. One can cite Prog. Polym. Sci. 2016, 58, 59.
L 254 check notation unit
L 462 perhaps good to include this in the introduction as well
Author Response
Reviewer #3:
The current work presents systematic data on novel polymer packaging compositions. The work has merit and needs only a few minor updates.
Reviewer’s Comment: L 190 please add a reference
Author’s Reply: Thanks for the comment. Proper reference added.
Reviewer’s Comment: L 207 intramolecular is likely incorrect as this means that only one type of molecule is involved. Later one intermolecular is used for two starch molecules. Intramolecular is vs additive but this is also intermolecular between starch molecule and additive functional group.
Author’s Reply: Thanks for the comment. It was revised.
Reviewer’s Comment: Ref 34. Please refer also to Free Volume Theory and its application in polymer engineering. One can cite Prog. Polym. Sci. 2016, 58, 59.
Author’s Reply: This is a great idea, we have revised and cited the awesome reference.
Reviewer’s Comment: L 254 check notation unit
Author’s Reply: Thanks for precious comment. Corrected in both text and figure.
Reviewer’s Comment: L 462 perhaps good to include this in the introduction as well
Author’s Reply: Dear reviewer, many thanks for your all your comments. You are correct, but we prefer to keep the introduction short and informative.
Round 2
Reviewer 1 Report
The authors have reflected all the correction as suggested, the manuscript improved significantly after revision. I suggest editor that manuscript can be accepted in present form.